# The Relationship between Nursing Practice Environment and Pressure Ulcer Care Quality in Portugal’s Long-Term Care Units

**DOI:** 10.3390/healthcare11121751

**Published:** 2023-06-14

**Authors:** Katia Furtado, Jaco Voorham, Paulo Infante, Anabela Afonso, Clara Morais, Pedro Lucas, Manuel Lopes

**Affiliations:** 1Out Patient Department, Hospital of Portalegre, Unidade Local de Saúde do Norte Alentejano, 7300-312 Portalegre, Portugal; 2Nursing Research, Innovation and Development Centre of Lisbon (CIDNUR), Escola Superior de Enfermagem de Lisboa, Av. Prof. Egas Moniz, 1600-096 Lisbon, Portugal; 3Comprehensive Health Research Centre (CHRC), Universidade de Évora, 7000-671 Évora, Portugal; 4DTIRS—Data to Insights Research Solutions, 1750-307 Lisboa, Portugal; jaco@dtirs.com; 5Research Center in Mathematics and Applications (CIMA), Instituto de Investigação e Formação Avançada (IIFA), Universidade de Évora, 7000-671 Évora, Portugal; 6Departamento de Matemática, Escola de Ciências e Tecnologia (ECT), Universidade de Évora, 7000-671 Évora, Portugal; 7Administração Regional de Saúde do Alentejo, Largo do Jardim do Paraíso, nº 1, 7000-864 Évora, Portugal; 8São João de Deus School of Nursing, Universidade de Évora, 7000-671 Évora, Portugal

**Keywords:** pressure ulcers, long-term care, nursing homes, work environment, quality of care, healing rates, Nursing Work-Revised Scale, risk factors, pressure injury, prevention

## Abstract

Background: The morbidity associated with ageing has contributed to an increase in the prevalence of Pressure Ulcers (PUs) in all care settings. The impact of these on people’s quality of life and the extent of the associated economic and social burden constitutes today, by their importance, a serious public health problem. This study aims to describe the nursing work environment in Portuguese long-term care (LTC) units and to assess how this environment relates to the quality of PU care. Methods: A longitudinal study among inpatients with PUs was conducted in LTC units. The Nursing Work Index-Revised Scale (NWI-R) was sent to all nurses in these units. Cox proportional hazard models were used to relate the satisfaction degree with the service (measured by the NWI-R-PT items) to the healing time of the PUs, adjusting for confounders. Results: A total of 165 of 451 invited nurses completed the NWI-R-PT. Most were women (74.6%) and had 1 to 5 years of professional experience. Less than half (38.4%) had education in wound care. Of the 88 patients identified with PUs, only 63 had their PU documented, highlighting the difficulties in updating electronic records. The results showed that the level of concordance with Q28 “Floating so that staffing is equalised among units” is strongly associated with a shorter PU healing time. Conclusion: A good distribution of nursing staff over the units will likely improve the quality of wound care. We found no evidence for possible associations with the questions on participation in policy decisions, salary level, or staffing educational development and their relationship with PUs healing times.

## 1. Introduction

Pressure ulcers, sometimes known as bedsores or pressure sores, are an injury that affects areas of the skin and underlying tissue. They are caused by pressure or pressure and shear [1] and are responsible for high morbidity and mortality rates [2,3]. There are various evidence-based preventative interventions for pressure ulcers, but their implementation in clinical practice is limited [4,5,6]. PUs are common in many care settings, with adverse health outcomes and high treatment costs [7]. These lesions are difficult to treat, and standard treatments do not have satisfactory healing rates [8]. It is, therefore, important to identify individuals with a higher risk of developing PUs to adopt the indicated preventive measures. Prevention is essential because PUs have devastating consequences on the individual’s quality of life, leading to more frequent hospitalisation and higher mortality [9,10,11,12].

According to the Organisation for Economic Cooperation and Development (OECD), there are, on average, five long-term care workers per 100 people aged 65 and over across 32 OECD countries, ranging from 12 in Norway and Sweden to less than one in Greece, Poland, and Portugal. The SARS-CoV-2 pandemic has exacerbated the need for higher staffing levels to replace sick or to isolate long-term care workers [13]. Nearly all OECD countries with available data have introduced measures (such as funding) to recruit long-term care professionals directly or indirectly. In Portugal, nurses have large workloads due to a shortage of nurses and double employment due to low wages [14]. The high complexity of patients and increased workload are often relevant barriers to implementing appropriate and timely prevention strategies [15,16].

Although PU prevention is a focus of the nursing care in long-term care units of Alentejo, acquired PUs continue to occur in these settings [17,18] eventually due to the complex nature of the patients’ health conditions [19]. Prevention work should be evidence-based and of a sufficient quality to inform and attain cost savings in PU treatment and to prevent suffering for patients [20]. Several evidence-based measures were recently implemented in the LTC units of Alentejo, including visits by a wound care specialist nurse, ongoing education sessions, and wound assessment via telemedicine. However, this did not result in a decrease in PU prevalence [21]. Across OECD countries, the observed prevalence of pressure ulcers in long-term care (LTC) units was 5.4%. The highest prevalence was observed in Spain (9.7%), Italy (9.9%), and Portugal (13.1%) [22].

A recent systematic review of pressure injury prevention among critically ill patients [23] reported that integrating several core components improved care processes and reduced pressure ulcer rates. Key features included simplifying and standardising pressure ulcer-specific interventions and documentation, multidisciplinary teams and leadership involvement, ongoing staff education, and sustained audit and feedback. A systematic review based on interventions in long-term care units found that PU rates decreased using computerised decision-making support systems, PU prevention programmes, repositioning or advanced cushions, or adding protein and energy supplements to diets [2]. Previous studies conducted in Portugal’s LTC units by our research group identified barriers to implementing evidence-based practice; specifically, low levels of knowledge among nurses about preventing pressure ulcers [24], inaccurate record keeping, and difficulties in contracting and retaining nurses in these units.

The role of organisational attributes is becoming increasingly important in ensuring that adequate staffing levels can be maintained despite nursing shortages. Studies conducted over the last ten years in “magnet” hospitals, which are so-called because they attract and retain professional nurses, have emphasised this point [25]. A positive nursing environment, where autonomy, adequate staff levels, control over practice, and good professional relationships are promoted as standards of good practice, can also improve nursing satisfaction and retain nursing staff [26,27,28]. In Portugal, this culture has been started by the creation of clinical research centres in private institutions and the National Health Service [29].

Besides this, the nursing practice environment can significantly impact patient outcomes by improving the quality of care and patient safety [27,30,31,32,33,34]. Mutair [35] concluded that PUs implementation interventions are worthwhile and can have long-term effects if carried out in a supportive and respectful environment. Involving staff in the implementation of updated policies and procedures and setting documentation requirements will reinforce new behaviours in a strong care culture [31,36,37,38]. Only a few studies have related the nursing work environment to patients’ wound care quality of care [35,39,40].

This study aims to describe the nursing work environment and to analyse how this environment relates to the quality of pressure ulcer care in LTC units in Alentejo, Portugal. It is part of a broad project to study barriers to the implementation of evidence-based practice related to wound care in LTC units, which is based on the premise that patient outcomes are influenced by how care services are organised and delivered. The organisation of care is a multidimensional concept influenced by knowledge, leadership, the work environment, and financial resources.

By identifying systematic barriers and facilitators for the adoption of evidence-based interventions in LTC units, we will contribute to the active management of obstacles and the use of positive approaches to facilitate the adoption of evidence-based practices related to the prevention of PUs [41,42,43].

## 2. Materials and Methods

Our research question is to understand how the LTC unit’s environment in Alentejo influences the healing rates of PUs. We looked at the associations between components of the Nursing Work Index-Revised Scale (NWI-R-PT) [44] and wound healing rates obtained from PUs identified in the 2018 prevalence study [21]. We focused on the institutionalised population in The National Network of Long-Term Care (RNCCI) [45]. There are four types of units with different goals: Units of Short-stay and complex cases (UC), Units of Medium-Stay and Rehabilitation (UMDR), Units of Long-Stay (ULDM), and Units of Palliative Care (UCP) [46].

### 2.1. Nursing Environment

#### 2.1.1. Study Design and Ethical Considerations

This is a quantitative, observational, descriptive, and cross-sectional study. We used an anonymous online survey between April and July 2021 among nurses from all inpatient long-term care units in Alentejo, Portugal. Formal authorisation for this survey was obtained from the Ethical Committee and Board of Administration of Health in Alentejo (protocol code 27/CE/2021). All participants signed an e-informed consent before completing the questionnaire and could withdraw from the study anytime.

Data were collected using a commercially available online survey tool (Google forms). The tool was hosted off-site from the server of the host academic institution, with the questionnaire design and analysis of the full data set being available to one named administrator (PI).

#### 2.1.2. Measurement Instrument

The Nursing Work Index was first developed by Kramer and Hafner in 1989 and contained 65 items [47]. In 2000, Aiken and Patrician revised it and presented a new version named the Nursing Work Index Revised (NWI-R) [48], with 57 items. Since then, the instrument has been widely used to measure the nursing practice environment and study its relationship with patient outcomes and job satisfaction [30,33,34,36,49,50]. In the present study, the validated Portuguese version of the NWI-R-PT was used [44].

The NWI-R is an instrument to assess the nursing environment in hospitals, primary care, long-term care, and nursing homes. It has several subscales: (a) staffing and resource adequacy, (b) nurse manager’s ability, (c) nurse participation and advancement, (d) nursing model for professional growth and quality assurance, and (e) collegial nurse–physician relationship [51]. The NWI-R-PT has 31 items, each measured on a 4-point Likert scale (1—strongly agree to 4—strongly disagree) and one open question for suggestions (Table 1). The questionnaire consists of three parts: Part A—Sociodemographic Characteristics; Part B—Characteristics of the nursing practice environment most valued by Nurses; Part C—Suggestions for improving practice. The results will be presented in this order.

The NWI-R-PT was selected for this study because all the items are relevant to our situation and take only five to ten minutes to complete. In addition to the original instrument, we included the following questions: gender, age, time working as a nurse, time working as a nurse in the institution, academic degree, duties in the institution, and work location.

A pilot study was performed with twenty-two nurses to ensure all questions were clear and appropriate and there was no need to change the instrument. These twenty-two questionnaires were included in the final sample.

#### 2.1.3. Data Collection

All 451 nurses working in LTC units in Alentejo were invited to participate in the study via e-mail by sending the invitation to the coordinator of each unit, who then distributed it to the individual nurses. No exclusion criteria were used.

Data were collected between 19 April and 22 July 2021, i.e., during the SARS-CoV-2 pandemic. The NWI-R-PT results were entered using the platform LimeSurvey (V. 3.25.17+210309). No information was collected on the nurses who did not participate in the study.

### 2.2. Wound Healing

#### 2.2.1. Design

This longitudinal study took place in LTC units from Alentejo, where inpatients with pressure ulcers were identified in February 2018 during the prevalence study [21]. At that point, 88 patients with Pus were identified. The time between wound detection and wound healing was measured for each patient. A wound was considered healed if the healing score did not exceed 15 points on the PUSH Tool Instrument [52,53] on the date of the PU assessment. Patients were censored if the PU had not healed by April 2020, when the patient died, or was lost to follow-up, whichever occurred first. Of the 88 patients identified with PUs, only 63 had records about their PUs. Most patients were in UMDR (59%), 22% in ULDM, and 18% in UC and 1% in UCP. For each patient, the most severe pressure ulcer (which affected the patient’s life most) was selected for follow-up to measure the time until healing. Only 30 patients had complete information for the selected PU.

#### 2.2.2. Measurement Instrument

The USA National Pressure Injury Advisory Panel [54] (NPUAP, 1997) developed the PUSH tool, which classifies PUs based on wound size, type of tissue present in the wound bed, and amount of exudate. There are other scales, such as the Bates–Jensen wound assessment tool [55,56] and the more recent Resvech 2.0 [57], with more evaluation parameters. However, the PUSH Tool is the selected scale available in the electronic system of LTC units in Portugal to help the continuous and consistent evaluation and monitoring of the PUs [58,59].

### 2.3. Statistical Analysis

The continuous variables were summarised using the mean (standard deviation), and categorical variables using each category’s frequency (percentage). The chi-square test of independence was used to test the existence of a significant relationship between time of professional experience in the current organisation and education in wound care. Cronbach’s alpha was used to assess the internal consistency of the six psychometric factors of the Portuguese version. To compare age by gender and worked hours by unit, we used the Wilcoxon Mann–Whitney test and the Kruskal–Wallis test, respectively, since the normality of these variables’ distribution was rejected by the Shapiro–Wilk test and the equality of the variances was not rejected by the Levene test.

The Goodman–Kruskal gamma correlation coefficient was used to measure the association between the level of agreement about organisational characteristics in the current workplace and the time of professional experience and professional experience in the current organisation.

We merged the information from the two sources to assess the relationship between the pre-selected set of NWI-R-PT items (predictors) and the time until pressure ulcer healing (outcome). The outcome was observed directly from the patients’ PUs, while the NWI-R-PT information was collected independently from the nurses working in the institutions where the patients were treated for their PUs. The common factor between the two data sources was the unit typology. Therefore, we aggregated the NWI-R-PT responses to median values per unit type (ULDM, UMDR, or UC). Since most nurses worked in several units, each nurse was allocated to the unit type where the nurse worked more hours monthly. In the case of a tie, random allocation was used. Bootstrap samples of nurses were generated, followed by aggregation of the questionnaire items at the typology unit level, and finally merged with the wound healing data.

A time-to-event design was used to estimate the association between risk factors and the individual predictors from the working environment and the time until wound healing.

Kaplan–Meier estimated survival curves were used to visualise the cumulative probability of wound healing over time by the PU severity grade and location. To compare the curves, we used the log-rank test or the Peto–Peto test, depending on if the survival curves do not cross or cross. The *p*-values were adjusted to correct for false discovery rates [60].

Cox models with a frailty component at the typology unit level were applied to the healing time of the PUs. A set of confounder candidates was generated from a previous study, being the most relevant risk factors for the occurrence and severity of PUs [21]: gender, grade of PU, location of PU, the origin of PU and existence of immobility, stroke, diabetes, dyslipidaemia, and rheumatological disease. Confounders were retained in the model if the effect size of the predictor changed by more than 20% after adding it to the previous model. This high cut-off was indicated due to the low sample size and the low number of outcomes. The confounder candidate that had the most impact on the coefficient was introduced first, followed by the one that had the most impact among the others and that independently changed the predictor coefficient by at least 20%, already in the presence of the first confounder. This algorithm stopped when a confounder candidate caused a change in coefficient below 20%. The predictors’ *p*-values were adjusted for multiple testing using the false discovery rate [60]. For each model, the assumption of proportional hazards was tested using Harrell’s test.

## 3. Results

### 3.1. Nursing Environment

#### 3.1.1. Sociodemographic Characteristics

A total of 165 (37%) nurses completed the NWI-R-PT questionnaire. Over half of the nurses worked in ULDM (51.4%), 29.7% in UMDR, 16.5% in UC, and 2.4% in UCP (Table 2). Most nurses worked in Portalegre (38.8%), followed by Beja (26.1%), and Évora (20.0%), and the remaining 15.1% worked in the Alentejo Coast region.

Although 75% of the recorded ages were between 21 and 36, the remaining 25% were between 36 and 68, comprising an extensive set of values with four outliers. Since these did not negatively influence the analyses, they were not removed from the sample. The number of working hours in March was missing for 27 nurses, which led to their exclusion from the analyses, resulting in a final sample of 138 nurses.

Most of the respondents were women (74.6%). The mean age of the nurses was 32.1 years, and 62.3% were at most 32 years (range 21 to 68 years). Less than half (38.4%) had specific education in wound care, which was usually a short-term course (64.2%) (Table 2).

More than half of the nurses (63.8%) did not have other employment (Table 3). Around half of the nurses (52.1%) had at most five years of professional experience, and 71.0% had been in their current organisation for less than six years. There is a significant relation between professional experience in the organisation and education in wound care (χ^2^(3) = 9.747, *p* = 0.021). Twenty-two per cent of the nurses with more than one year of experience in the organisation had wound training, while this was the case in 6% of the nurses with shorter experience.

#### 3.1.2. Work Environment

In their current workplace, 51% of the nurses agreed that the “supervisory staff is supportive of nurses” (Q2), 49% agreed that “each nursing unit determines its policies and procedures” (Q29), and 41% agreed with “floating so that staffing is equalised among units” (Q28). Around 40% of the nurses did not agree or disagree that “nursing staff is supported in pursuing degrees in nursing” (Q17). The opinion of nurses on “career in-service/continuing education programs for nurses” (Q4) and on the “opportunities for staff nurses to participate in policy decisions” (Q5) was almost equally distributed between disagreement, indifference, and agreement. There was no clear agreement on “enough staff to get the work done” (Q12) in the current workplace: 2 out of 5 nurses agree, and 2 out of 5 disagree. There were weak significant negative correlations between the level of agreement about “enough staff to get the work done” (Q12) and time of professional experience (g = −0.270, *p* = 0.003), time of professional experience in the current organisation (g = −0.254, *p* = 0.010), and “floating, so that staffing is equalised among units” (Q28) and time of professional experience (g = −0.184, *p* = 0.036).

In question 3, “a satisfactory salary”, the percentage of nurses who disagree is slightly higher (32%) than the ones who agree (29%), so the salary is generally considered unsatisfactory. Question 8, “enough registered nurses on staff to provide quality patient care” relates to the ratio of nurses-per-patient, and most nurses consider the ratio insufficient. However, there are enough experienced nurses working who “know” the Institution (Q30). Question 25, “regular, permanently assigned staff nurses never to float to another unit”, shows a conflict between nurses. Only question 28 is also related to “floating, so that staffing is equalised among units” and shows that nurses tend to consider positive floating between services during some shifts (40 against 28%). Concerning question 29, “each nursing unit determines its own policies and procedures”, most nurses assumed a neutral position or agreed with it.

#### 3.1.3. Suggestions for Improving Practice

The last part of the NWI-R-PT questionnaire is an open question for suggestions or other important topics that should be addressed. We have analysed the content of this section by relating the 46 issues raised by nurses with questions asked previously in the NWI-R-PT. Table 5 shows the most frequently mentioned issues.

We selected the five most frequently mentioned questions in the NWI-R-PT open-ended question (Table 6) as the candidate predictors of the quality of care provided in healing pressure ulcers.

### 3.2. Wound Healing

The sample of 63 patients had a few more female patients (*n* = 34). Almost all patients were elderly people (75% of the patients were over 75-years-old), with a mean age of 70.8 years and a standard deviation of 11.8 years. More than 9 in 10 patients had at least two risk factors (92%). The most frequent risk factors were arterial hypertension (*n* = 36), immobility (*n* = 25), stroke (*n* = 23), diabetes (*n* = 15), dyslipidaemia (*n* = 13), cardiac insufficiency (*n* = 13), and rheumatological diseases (*n* = 14).

The most frequent locations of the PUs were the sacrococcygeal region (33%), trochanter (28%), and calcaneus (25%). Concerning the origin of the PU, the highest percentage comes from the hospital (37%), followed by home (32%), and RNCCI (21%). More than half of the patients had a grade 4 PU (56%), 26% had a grade 3 PU, 10% had a grade 2 PU, and 8% had a grade 1 PU. Only 25 PUs (40,3%) healed during the patient stay in these units.

Figure 1 shows the estimated Kaplan–Meier healing curves for the different PU grades, showing the importance of this factor in PU healing (*p* < 0.001). Wounds with grade 4 needed, on average, more time to heal than wounds of grade 1 (adjusted *p* < 0.001, HR = 244.5, HR CI95% = (24.2, 2466.7)), grade 2 (adjusted *p* < 0.001, HR = 9.5, HR CI95% = (2.3, 39.4)), and even when compared to wounds of grade 3 (adjusted *p* = 0.024, HR = 3.9, HR CI95% = (1.4, 10.5)). Grade 1 wounds heal, on average, faster than grade 3 wounds (adjusted *p* < 0.001, HR = 63.2, HR CI95% = (6.6, 602.4)).

There were no statistically significant differences in time until wound healing between the origins of the PU (*p* = 0.300). However, the location of the PU is a relevant factor in the healing. Figure 2 shows the Kaplan–Meier healing curves for the different locations of the PU. A pressure ulcer located at the calcaneus needs, on average, less time to heal than a PU located in the sacrococcygeal region (adjusted *p* = 0.030, HR = 3.0, HR CI95% = (1.1, 8.4)) and at the trochanter (adjusted *p* = 0.005, HR = 22.0, HR CI95% = (2.7, 178.3)).

### 3.3. Relationship between Work Environment and Wound Healing

The results of the Cox regression models for the five selected predictor candidates are presented in Table 7. A higher concordance of the nurses with “floating, so that staffing is equalised among units” (Q28) is strongly and significantly associated with a shorter time until PU healing. There is an indication that the higher the concordance of the nurses with “nursing staff is supported in pursuing degrees in nursing” (Q17), the longer the time until the PU healing is, although this did not reach statistical significance. Likewise, a higher concordance of the nurses with the “opportunity for staff nurses to participate in policy decisions” (Q5) or “satisfactory pay” (Q3) shows to be related to shorter healing times, although not reaching statistical significance. The item “sufficient staffing for the provision of care” (Q12) seems unrelated to PU healing time.

## 4. Discussion

In this study, we intended to evaluate if the work environment is associated with wound care quality in LTC units. The results showed that the level of concordance with Q28 “Floating so that staffing is equalised among units” is strongly associated with a shorter PU healing time. We found no evidence for the possible associations with questions on participation in policy decisions, salary level, staffing educational development, and PUs healing times. Although the literature on this topic is scarce, previous studies show that positive nursing work environments are related to better patient outcomes, including implementing evidence-based practices [31,61,62]. Although the questionnaire results showed these are relevant issues for the nurses, they try to ensure better care for patients with PUs, such as floating between units. Previous reviews have identified no consistent relationship between staffing levels and nurse-sensitive patient outcomes [63,64]. Nursing care models are complex and vary with the level of education, experience, nursing care hours required per patient, individual nurse characteristics, and patient care needs, among other factors [65].

The NWI-R-PT was answered by 38% of the nurses working in the LTC units of Alentejo during the SARS-CoV-2 pandemic. Since 2020, nurse turnover has increased globally [66]. In this period, public hospitals offered better financial conditions for nurses in response to the growing needs. This caused a fluctuation in the number of nurses working in these units, hampered the data collection process, and likely affected the response rate.

Most nurses were young, 62% were younger than 32, and most had less than five years of professional experience. In Portugal, it is common for nurses to have more than one job due to low salaries, but in our sample, more than half of the nurses (63.8%) did not have other employment but worked in more than one unit typology. This could mean they had the conditions to commit to the values and mission of the organisation.

A supportive work environment is essential for nurses’ quality of working life and improved patient outcomes [36]. Nurses’ salaries and careers are the main issues that need to improve in long-term care units to retain professionals and create a positive culture towards better practices.

In our sample, education in wound care awareness increases with experience, but knowledge is insufficient [24,67]. Wound care management is an important aspect of daily care for patients in long-term care units. Surgical wounds care and pressure ulcers represent 17 and 22% of the cases referred to these units, respectively [18].

Regarding the units’ capacity, the nurses stressed the need for adequate labour conditions, including a good nurse/patient ratio, to cover the needs of patients. Previous studies showed that most patients in these units suffer from multimorbidity and have physical limitations [68]. Inadequate staffing levels are associated with nurses’ job dissatisfaction and burnout, which often increases turnover rates [26,69].

The presence of experienced nurses who know the organisation, and do not merely deliver their nursing hours, is relevant for promoting a good environment, as observed in other studies [40,70]. Aiken shows that reducing nurses’ workloads by adding additional nurses alone may have little consequence; nursing characteristics need to be considered in combination [61].

Nurses with authority and autonomy to make decisions and control patient care resources can establish a better relationship with physicians [26,48]. We observed that nurses consider “floating between services during some shifts” relevant. Most nurses assumed a neutral position or agreed with the question, “each nursing unit determines its own policies and procedures”. To the open question of the questionnaire, nurses answered that they were excluded from decision-making regarding patient issues. A stronger autonomy with control over the working environment and organisational support leads to better relations between physicians and nurses [32,71,72].

Concerning wound healing, from the 88 patients identified with pressure ulcers, only 63 had their PU well-documented; of these, only 30 had complete records. This fact highlighted the poor quality of record-keeping in RNCCI and was also mentioned by the nurses in the NWI-R-PT questionnaire as an issue that needs improvement. Gunningberg found the same results when comparing the accuracy of recorded PU prevalence data and prevention data [73]. Accurate and complete data are required to help diagnose, prevent, and treat PUs and ensure the quality of care in LTC units [74,75,76,77,78]. Finally, the presence of two different electronic documentation systems in LTC units may be causing deficits in pressure ulcer reporting [79].

Our study sample showed an elderly population with multimorbidity, where immobility is a strong risk factor for poor healing [80,81,82,83]. The high percentages of grade 3 (26%) and 4 (57%) PUs are not aligned with other studies conducted in LTC units [2,84,85,86], indicating limited prevention interventions [24].

Although there is no established way to organise care for pressure ulcer prevention and treatment, there are examples of provider-orientated interventions by introducing wound care champions figures, evidence-based care pathways, and improvements in nursing documentation [78,79].

### Limitations

The results of this study should be interpreted with caution since it used a cross-sectional design limited to the long-term care units of a region of Portugal. Future research with larger samples and in different settings is needed to provide more information on the nursing work environment and its association with pressure ulcer healing.

The number of pressure ulcers followed was low, and the records were inaccurate, resulting in limited statistical power. This could be a potential reason for the weak correlation between the nursing work environment and PUs’ quality of care. Besides this, the information on the nurses’ working environment and wound healing came from two separate studies, and most of the nurses who answered the NWI-R-PT were there for no more than five years.

The available confounder candidates were a limited set of risk factors for PUs, and we could not take other possible confounders into account, such as incontinence, lack of sensory perception, obesity, poor nutrition and hydration, heart failure, renal disease, and mental illnesses such as Alzheimer’s disease.

## 5. Conclusions

In general, nurses are satisfied with their work environment in long-term care units in Alentejo. We found low-quality and inaccurate nursing documentation related to wound care and observed low healing rates. Age, immobility, and the presence of more than two risk factors impaired healing. A good distribution of nursing staff over the units might improve the quality of wound care, but the other studied work environment factors did not impact the quality of care.

Since the current shortage of nurses in long-term care units is a global problem, there is an urgent need to change the system to create more supportive nurse work environments to retain nurses. Some strategies indicated by nurses from these units are offering better salaries, professional progression, and involving nurses in institution projects and top decisions.

These findings provide important lessons for administrators and supervisors interested in promoting quality improvement in long-term care units. Documenting the nursing process is essential for supporting healthcare decisions, improving patient care, and ensuring patient safety. This study provides information to guide policies to improve the quality of nursing documentation systems in LTC units.

Bridging the gap between practice and education is also important for enhancing nursing competencies to ensure they will be ready to meet the demands of an ageing population with multimorbidity. The prevention of PUs should be considered a mandatory part of undergraduate nursing programs.

This is the first study in Portugal concerning the nursing work environment in LTC units and the quality of PUs care. More studies are needed, with larger samples to allow benchmarking and help decision-making.

## Figures and Tables

**Figure 1 healthcare-11-01751-f001:**
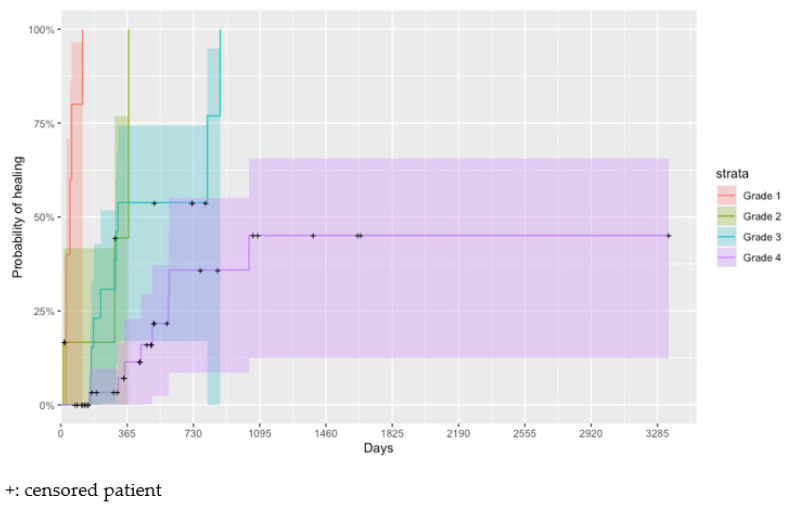
Estimated Kaplan–Meier cumulative probability of wound healing (95% confidence interval) by PU grade.

**Figure 2 healthcare-11-01751-f002:**
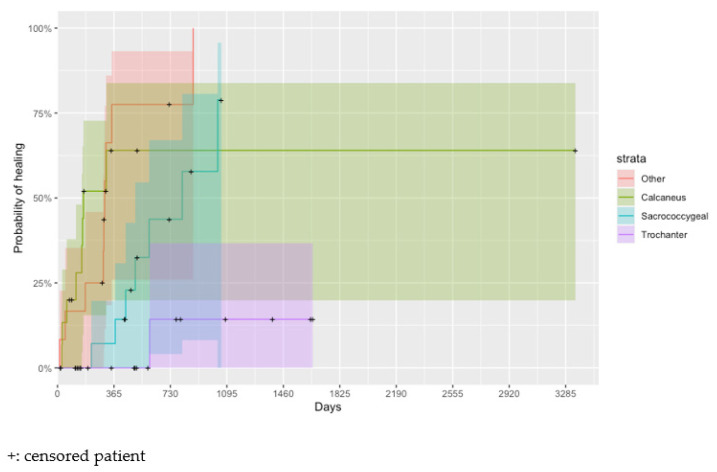
Estimated Kaplan–Meier cumulative probability of wound healing (95% confidence interval) by the location of the wound.

**Table 1 healthcare-11-01751-t001:** NWI-R-PT components.

Component	Item	Cronbach’s Alpha		Component	Item	Cronbach’s Alpha	
Management support	2	0.75	0.86	Collegial Nurse-Physician relations	1	0.63	0.77
	6	0.5			15	0.7	
	7	0.63			20	0.74	
	9	0.74			24	0.92	
	10	0.66		Endowments	8	0.81	0.65
	13	0.68			12	0.9	
	18	0.7			28	0.26	
	23	0.65		Organisation of nursing care	11	0.67	0.59
Professional development	3	0.45	0.81		14	0.37	
	4	0.7			25	0.35	
	5	0.7			26	0.63	
	16	0.71					
	17	0.63					
	19	0.62					
Fundamentals of nursing	21	0.57	0.69				
	22	0.55					
	27	0.54					
	29	0.39					
	30	0.6					
	31	0.55					

**Table 2 healthcare-11-01751-t002:** Response by unit.

Long-Term Care Unit Type	Population	Sample Number (%)
ULDM (long stay)	206	31 (15.0)
UMDR (medium stay and rehabilitation)	122	69 (56.6)
UC (short stay and complex cases)	102	38 (37.3)
UCP (palliative care)	21	0 (0.0)
Total	451	138 (30.6)

**Table 3 healthcare-11-01751-t003:** Sample characteristics.

Variable	Category	*n* (%)
Gender	Female	103 (74.6)
	Male	35 (25.4)
Education	Graduate	128 (92.7)
	Master	10 (7.3)
Education in Wound Care	No	85 (61.6)
	Yes	53 (38.4)
Type of wound training	Short-term course	34 (64.2)
	Postgraduation	19 (35.8)
Professional Experience	<1 year	26 (18.8)
	1–5 years	46 (33.3)
	6–10 years	34 (24.6)
	11–20 years	17 (12.3)
	>20 years	15 (11.0)
Professional Experience in the organisation	<1 year	38 (27.5)
	1–5 years	60 (43.5)
	6–10 years	25 (18.1)
	11–20 years	15 (10.9)
Other employment	No	88 (63.8)
	Yes	50 (36.2)

No significant differences were found in the number of hours worked between the unit types (*p* = 0.205; Table 4).

**Table 4 healthcare-11-01751-t004:** Number of hours that nurses worked in each unit in one month.

Unit	*n* Nurses	Hours Worked	
		Median (IQR)	Range
ULDM	109	80 (48–140)	8–190
UC	35	120 (72–148)	8–192
UMDR	63	76 (46–140)	8–192
UCP	5	40 (12–140)	12–192

**Table 5 healthcare-11-01751-t005:** Answers to part C of the second section (Suggestions).

Education in Wound Care (Q17)	No opportunities to attend course.
	No nurses with knowledge, competence, and experience (lots of young nurses)
Units Capacity (Q12), (Q28)	Need an adequate nurse/patient ratio and to reduce workload.
	Inadequate labour conditions for the real needs of patients (few nurses, constrains in the equipment)
Conflicts between financial income and fair salaries (Q3) (Q28)	The salary in long-term care units is lower than in the NHS
	Need more flexibility in schedules and holidays.
Nurse–Physician relationship (Q5)	No support from physicians.
	Nurses are excluded from decision-making in issues related with patients.
Structural and political changes are necessary in long-term care units (Q5)	Political and organisational changes are needed.
Improvements in monthly documentation in GESTCare^®^ platform (Q5)	The electronic record system needs an update.

**Table 6 healthcare-11-01751-t006:** Selected questions.

Question Number	Description
Q3	A satisfactory salary.
Q5	Opportunity for staff nurses to participate in policy decisions.
Q12	Enough staff to get the work done.
Q17	Nursing staff is supported in pursuing degrees in nursing.
Q28	Floating, so that staffing is equalizes among units.

**Table 7 healthcare-11-01751-t007:** Results from the prediction models of nursing work environment factors on time until PU healing.

Predictor	HR (95% CI)	*p*-Value *	Confounders
Q28	0.09 (0.02, 0.50)	0.030	PU Source PU Local
Q17	21.54 (1.46, 317.12)	0.100	PU Local
Q12	0.48 (0.05, 4.39)	0.520	PU Grade Immobility
Q5	0.04 (0.00, 0.92)	0.132	PU Grade
Q3	0.20 (0.03, 1.34)	0.194	PU Grade Immobility

* Adjusted for multiple testing.

## Data Availability

Data are unavailable due to privacy.

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
