# Peer review of "The Relationship between Nursing Practice Environment and Pressure Ulcer Care Quality in Portugal’s Long-Term Care Units"

_healthcare, 2023, doi:10.3390/healthcare11121751_

Round 1
Reviewer 1 Report
Dera Authors:
Here are a few suggestions for improving the abstract:
- Provide more context: The abstract starts by stating that pressure ulcers (PUs) are responsible for high morbidity and mortality, but it doesn't provide any additional information about the scope or impact of the problem. Adding a sentence or two that gives some context would help readers understand why this study is important.
- Be more specific about the methods: The abstract mentions that the study looked at associations between the nursing work environment and wound healing rates, but it doesn't provide much detail about the methods used to collect and analyze data. Adding a sentence or two that describes the study design and methods would make the abstract more informative.
- Use clearer language: Some of the sentences in the abstract are a bit convoluted, which can make it harder for readers to understand the main findings. Using simpler language and shorter sentences can help make the abstract more accessible.
To improve the introduction, a few changes can be made:
- The first sentence can be revised to provide a more precise definition of pressure ulcers and clarify the problem’s significance.
- The second paragraph can be reorganized to emphasize the staffing challenges faced by Portugal and how it relates to pressure ulcer prevalence.
- In the third paragraph, the sentence about the prevalence of pressure ulcers across OECD countries can be moved to the end of the paragraph to better connect it with the following discussion of pressure ulcers in Alentejo, Portugal.
- The fourth paragraph can be revised to provide more details on the recent evidence-based measures implemented in Alentejo and why they may have been unsuccessful.
- The fifth paragraph can be rephrased to better integrate the discussion of previous research on pressure ulcer prevention and the role of organizational attributes in improving nursing satisfaction and patient outcomes.
- The last paragraph can be expanded to provide more details on the specific research questions and objectives of the study.
here are some suggestions to revise the Materials and Methods section:
- Provide a clearer research question and hypothesis. This will help readers understand the purpose of the study and how the methods were designed to answer the research question.
- Provide more detailed information about the sampling strategy. For example, how were the nurses selected to participate in the study? What were the inclusion and exclusion criteria?
- Clarify the methods used to measure wound healing rates. What was the PUSH Tool Instrument, and how was it used to assess wound healing?
- Provide more detailed information about the statistical analysis. For example, what specific statistical tests were used to analyze the data? What were the assumptions of these tests, and how were they verified?
- Consider revising the section headings to make them more descriptive and informative. For example, instead of "2.1 Nursing Environment," consider "2.1.1 Sampling Strategy" and "2.1.2 Measurement Instrument."
- Consider adding a subsection that describes the data cleaning and preparation process. For example, how were missing data and outliers handled?
- Provide more detailed information about the ethical considerations involved in the study. For example, what steps were taken to ensure participant confidentiality and informed consent?
- Consider adding a power analysis to justify the sample size used in the study. This will help readers understand the statistical power of the study and how well the sample size was able to detect significant associations.
- Provide a more detailed explanation of the variables and measures used in the study. For example, what specific items were included in each subscale of the Nursing Work Index-Revised Scale?
To improve the Results section, the author could consider the following suggestions:
- Provide more context: The author should provide more background information about the study's purpose, research questions, and hypotheses to help readers better understand the study's results.
- Organize the results: The author should organize the results section in a logical and coherent manner, grouping related findings together, and using subheadings to make it easier for readers to navigate the section.
- Provide more detailed information: The author should provide more detailed information about the results, including statistical analyses, effect sizes, and confidence intervals, to help readers evaluate the significance of the findings.
Here are some suggestions on how to improve the discussion section:
- Start with a clear and concise summary of the main findings of the study, followed by a brief discussion of how these findings relate to previous research in the field.
- Provide a more detailed analysis of the results, including a discussion of any unexpected findings or limitations of the study. For example, you could discuss any potential confounding variables that may have affected the results or any difficulties encountered during data collection.
- Draw connections between the results and the larger implications for practice or policy. This could include discussing how the results may inform future interventions or changes in healthcare delivery.
- Consider the relevance of the study's findings beyond the specific context of the study. This could include discussing how the findings relate to broader issues in healthcare or nursing.
- Provide recommendations for future research, including any areas where additional studies are needed or where the current study's limitations may be addressed in future research.
- Avoid overly technical language or jargon that may be difficult for readers to understand. Use clear and concise language, and define any technical terms or acronyms used in the discussion.
The conclusions section seems well-written and summarizes the key findings of the study. However, there are some ways it could be improved:
- Provide more specific recommendations: While the conclusion mentions the need to change the system to create more supportive nurse work environments, it would be helpful to provide more specific recommendations for how this can be achieved. For example, the study could suggest increasing the number of nursing staff in long-term care units or providing additional resources for wound care.
- Discuss limitations: The conclusion could also include a discussion of the limitations of the study. This would provide readers with a better understanding of the scope and generalizability of the findings.
- Mention future research directions: The conclusion could also mention potential areas for future research. For example, the study could suggest investigating the effectiveness of specific interventions aimed at improving nurse work environments in long-term care units.
Overall, the conclusion section provides a good summary of the study's findings, but could be improved by including more specific recommendations, discussing limitations, and mentioning future research directions.
The sentences are well-structured, clear and concise and use appropriate academic vocabulary. The language used demonstrates a good level of proficiency in English writing.
Author Response
Dear Reviewer,
Thank you for your attention and time spent reading our article. We really appreciate your generosity. We have considered your pertinent suggestions. To facilitate, we have written what we have changed in the article, bellow after each of your suggestion.
Regards
Dear Authors:
Here are a few suggestions for improving the abstract:
- Provide more context: The abstract starts by stating that pressure ulcers (PUs) are responsible for high morbidity and mortality, but it doesn't provide any additional information about the scope or impact of the problem. Adding a sentence or two that gives some context would help readers understand why this study is important.
We have added the following: The morbidity associated with ageing has contributed to an increase in the prevalence of Pressure Ulcers (PU) in all care settings. The impact of these on people's quality of life and the extent of the associated economic and social burden constitutes today, by their importance, a serious public health problem.
- Be more specific about the methods: The abstract mentions that the study looked at associations between the nursing work environment and wound healing rates, but it doesn't provide much detail about the methods used to collect and analyse data. Adding a sentence or two that describes the study design and methods would make the abstract more informative.
We have reorganised this section:
Methods: A longitudinal study among inpatients with PUs was done in LTC units. The Nursing Work Index-Revised Scale (NWI-R) was applied to a sample of nurses working in these units. Cox proportional hazard models were used to relate the satisfaction degree with the service (measured by NWI-R items) to the healing time of the PUs, adjusting for confounders.
- Use clearer language: Some of the sentences in the abstract are a bit convoluted, which can make it harder for readers to understand the main findings. Using simpler language and shorter sentences can help make the abstract more accessible.
We have reorganised the text.
To improve the introduction, a few changes can be made:
- The first sentence can be revised to provide a more precise definition of pressure ulcers and clarify the problem’s significance.
We have clarified the definition of PUs.
We have added more information about the dimension of the problem;
These lesions are difficult to treat and standard treatments do not have satisfactory healing rates (Kuffler, 2015). It is, therefore, important to identify individuals with a higher risk of developing PUs to adopt indicated preventive measures. Prevention is essential because PUs have devastating consequences on the individual's quality of life, leading to more frequent hospitalization and higher mortality (Goodall et al., 2020; Lyder, 2011; Mallah et al., 2015; Moore, Z.; Cowma, 2009).
- The second paragraph can be reorganized to emphasize the staffing challenges faced by Portugal and how it relates to pressure ulcer prevalence.
We have added:
In Portugal, nurses have large workloads due to a shortage of nurses and double employment due to low wages. The high complexity of patients and increased workload are often relevant barriers to the implementation of appropriate and timely prevention strategies (Coyer et al., 2019; Lin et al., 2020).
- In the third paragraph, the sentence about the prevalence of pressure ulcers across OECD countries can be moved to the end of the paragraph to better connect it with the following discussion of pressure ulcers in Alentejo, Portugal.
We have changed as suggested.
- The fourth paragraph can be revised to provide more details on the recent evidence-based measures implemented in Alentejo and why they may have been unsuccessful.
We have added:
Previous studies done in Portugal’s LTC units by our research group identified barriers to the implementation of evidence-based practice, specifically, low levels of knowledge among nurses about the prevention of pressure ulcers (Furtado et al., 2022), inaccurate record keeping, and difficulties in contracting and retaining nurses in these units.
- The fifth paragraph can be rephrased to better integrate the discussion of previous research on pressure ulcer prevention and the role of organizational attributes in improving nursing satisfaction and patient outcomes.
We have changed as suggested:
Mutair (Al Mutair et al., 2021) concluded that PUs implementation interventions are worth the effort and can have long term effects if carried out in supportive and respectful environment. Involving staff with implementation of updated policy and procedures and setting documentation requirements will reinforce new behaviors in a strong care culture (Aiken et al., 2018; Akhu-Zaheya et al., 2018; Kanai-Pak et al., 2008; Mihdawi et al., 2020).
- The last paragraph can be expanded to provide more details on the specific research questions and objectives of the study.
We have added:
By identifying systematic barriers and facilitators for the adoption of evidence-based interventions in LTC units, we will contribute to the active management of obstacles and in the use of positive approaches to facilitate the adoption of evidence-based practices related to the prevention of PUs (Craig et al., 2019; Ma et al., 2020; Skivington et al., 2021).
here are some suggestions to revise the Materials and Methods section:
- Provide a clearer research question and hypothesis. This will help readers understand the purpose of the study and how the methods were designed to answer the research question.
We have added:
Our research question is to understand how LTC unit’s environment in Alentejo, influences the healing rates of PUs.
- Provide more detailed information about the sampling strategy. For example, how were the nurses selected to participate in the study? What were the inclusion and exclusion criteria?
We have added in the section 2.1.3:
All 451 nurses working in LTC units in Alentejo were invited to participate in the study via e-mail, by sending the invitation to the coordinator of each unit, who then distributed it to the individual nurses. No exclusion criteria were used.
- Clarify the methods used to measure wound healing rates. What was the PUSH Tool Instrument, and how was it used to assess wound healing?
We have added a section on this topic:
Measurement Instrument:
The USA National Pressure Injury Advisory Panel (NPUAP, 1997) developed the PUSH tool, which classifies PUs based on wound size, type of tissue present in the wound bed and amount of exudate. There are other scales, such as the Bates-Jensen wound assessment tool (Bates-Jensen, 2005) and the more recent Resvech 2.0 (Restrepo-Medrano & Soriano, 2011), with more evaluation parameters. However, the PUSH Tool is the selected scale available in the electronic system of LTC units in Portugal, to help continuous and consistent evaluation and monitoring of the PUs.
- Provide more detailed information about the statistical analysis. For example, what specific statistical tests were used to analyze the data? What were the assumptions of these tests, and how were they verified?
We added in the methods the reference to the chi-square test and the Goodman-Kruskal correlation coefficient. We have tested normality and homogeneity for non-parametric tests.
We clarified when the log-rank test and the Peto&Peto test were used.
In the case of Cox models, since we have no continuous variables, only the assumption "hazards are proportional (PH)" was evaluated, as written at the end of the section "the assumption of proportional hazards was tested using Harrell's test."
- Consider revising the section headings to make them more descriptive and informative. For example, instead of "2.1 Nursing Environment," consider "2.1.1 Sampling Strategy" and "2.1.2 Measurement Instrument."
We have changed 2.1.2 to Measurement Instrument.
We have added missing information on the sampling strategy in the point of Study design and Data collection.
- Consider adding a subsection that describes the data cleaning and preparation process. For example, how were missing data and outliers handled?
We have added in the results section:
We have identified 4 outliers in the age of nurses. Although 75% of the recorded ages were between 21 and 36, the remaining 25% of ages fell between 36 and 68, comprising an extensive set of values with four outliers. Since this data did not negatively influence the process they were not removed.
We added to the text:
“The number of working hours in March was missing for 27 nurses, which led to their exclusion from the analyses, resulting in a final sample of 138 nurses.”.
- Provide more detailed information about the ethical considerations involved in the study. For example, what steps were taken to ensure participant confidentiality and informed consent?
We have added:
Data were collected using a commercially available online survey tool (google forms). The tool was hosted off-site from the server of the host academic institution, with the questionnaire design and analysis of the full data set being available to one named administrator (PI).
- Consider adding a power analysis to justify the sample size used in the study. This will help readers understand the statistical power of the study and how well the sample size was able to detect significant associations.
The sample size of this study was not defined based on statistical power. Instead, we used a convenience sample comprised of all available patients with PUs whose wound healing was recorded.
- Provide a more detailed explanation of the variables and measures used in the study. For example, what specific items were included in each subscale of the Nursing Work Index-Revised Scale?
We have added the table below:
Table 1 – NWI-R-PT components
To improve the Results section, the author could consider the following suggestions:
- Provide more context: The author should provide more background information about the study's purpose, research questions, and hypotheses to help readers better understand the study's results.
We have added this information in the previous sections.
- Organize the results: The author should organize the results section in a logical and coherent manner, grouping related findings together, and using subheadings to make it easier for readers to navigate the section.
We have added subtitles.
- Provide more detailed information: The author should provide more detailed information about the results, including statistical analyses, effect sizes, and confidence intervals, to help readers evaluate the significance of the findings.
We have added the suggested information.
Here are some suggestions on how to improve the discussion section:
- Start with a clear and concise summary of the main findings of the study, followed by a brief discussion of how these findings relate to previous research in the field.
We have added this information:
Although the literature on this specific topic is scarce, previous studies show that positive nursing work environments are related to better patient outcomes, including implementation of evidence-based practices.
- Provide a more detailed analysis of the results, including a discussion of any unexpected findings or limitations of the study. For example, you could discuss any potential confounding variables that may have affected the results or any difficulties encountered during data collection.
We have added information on this.
- Draw connections between the results and the larger implications for practice or policy. This could include discussing how the results may inform future interventions or changes in healthcare delivery.
We have added:
Documenting the nursing process is essential for supporting healthcare decisions, improving patient care and ensuring patient safety. This study provides information to guide policies to improve and identify effective strategies and actions to enhance the quality of nursing documentation systems in LTC units.
- Consider the relevance of the study's findings beyond the specific context of the study. This could include discussing how the findings relate to broader issues in healthcare or nursing.
We have added:
Bridging the gap between practice and education is important for enhancing nursing competencies to ensure that they will be ready to meet the demands of an ageing population with multimorbidity. Prevention of PUs should be considered a mandatory part of the undergraduate nursing programmes.
- Provide recommendations for future research, including any areas where additional studies are needed or where the current study's limitations may be addressed in future research.
We have added information on this in the conclusion section.
- Avoid overly technical language or jargon that may be difficult for readers to understand. Use clear and concise language, and define any technical terms or acronyms used in the discussion.
We have changed some of the language to improve readability.
The conclusions section seems well-written and summarizes the key findings of the study. However, there are some ways it could be improved:
- Provide more specific recommendations: While the conclusion mentions the need to change the system to create more supportive nurse work environments, it would be helpful to provide more specific recommendations for how this can be achieved. For example, the study could suggest increasing the number of nursing staff in long-term care units or providing additional resources for wound care.
We have added information on this topic: offer better salaries and professional progression.
- Discuss limitations: The conclusion could also include a discussion of the limitations of the study. This would provide readers with a better understanding of the scope and generalizability of the findings.
We have added:
This is the first study done in Portugal concerning the nursing work environment and the quality of PU care. More studies are needed, with larger samples to allow benchmarking and help decision making.
- Mention future research directions: The conclusion could also mention potential areas for future research. For example, the study could suggest investigating the effectiveness of specific interventions aimed at improving nurse work environments in long-term care units.
We have suggested to replicate the study with larger samples.
Overall, the conclusion section provides a good summary of the study's findings, but could be improved by including more specific recommendations, discussing limitations, and mentioning future research directions.
Comments on the Quality of English Language
The sentences are well-structured, clear and concise and use appropriate academic vocabulary. The language used demonstrates a good level of proficiency in English writing.
For more details please see the revised version manuscript.

Reviewer 2 Report
Dear authors. Thank you for your manuscript. Here are some comments, after reading it:
Line 60 - how many units in Alentejo were involved? Why these units?
Lines 71-72 - magnet hospitals are referred. Are there magnet hospitals in your country? Other management criteria could be addressed.
Design section - you refer to a prevalence study taken on February 2018. You identified 88 PU. Then you say the time of PU healing. When was healing measured? How did you monitor these PU from 2018 (prevalence study) and 2020? The 63 patients you mentioned are new records.
Paragraph line 145 - you have 2 types of sources: a prevalence study in 2018 and an assessment by NWI-R-PT in 2020. How can you merge these 2 sources? Were the nurses working in those units the same in 2020 and 2018? The PU in 2018 were the same as in 2020? Were there healing PUs?
Line 153 - how was randomization made?
Table 4 - missing a headline with categories and themes emerged by the open question. Please specify how have you analyzed the content from this open question. Why ask this question if the content doesn't value the text?
Discussion - on your results, you report 38.4% of nurses with Education in Wound Care. Did you analyse if the PUs on the patients where those nurses worked had a better outcome? Did they have a better healing time?
Kind regards,
Minor English corrections are required.
Author Response
Dear Reviewer,
Thank you for your attention and time spent reading our article. We really appreciate your generosity. We have considered your pertinent suggestions. To facilitate, we have written what we have changed in the article, bellow after each of your suggestion.
Regards
Dear authors. Thank you for your manuscript. Here are some comments, after reading it:
Line 60 - how many units in Alentejo were involved? Why these units?
“The wound healing study took place in LTC units from Alentejo where inpatients with pressure ulcers were identified in February 2018, during the prevalence study” . We have changed the sentence to make it clear, in the material and methods section.
Lines 71-72 - magnet hospitals are referred. Are there magnet hospitals in your country? Other management criteria could be addressed.
We have added:
In Portugal, this culture has been started by the creation of clinical research centers in private institutions and the National Health Service.
Design section - you refer to a prevalence study taken on February 2018. You identified 88 PU. Then you say the time of PU healing. When was healing measured? How did you monitor these PU from 2018 (prevalence study) and 2020? The 63 patients you mentioned are new records.
We have identified 88 patients in the prevalence study. From these, only 63 had records in the nursing electronic documentation system. We have followed them until healing. We have clarified the sentence in the text.
Paragraph line 145 - you have 2 types of sources: a prevalence study in 2018 and an assessment by NWI-R-PT in 2020. How can you merge these 2 sources? Were the nurses working in those units the same in 2020 and 2018? The PU in 2018 were the same as in 2020? Were there healing PUs?
Most of the nurses who answered the NWI-R.PT were working in these units during the prevalence study done in 2018 because more than half had 5 years of experience in the institution. We have explained this in the limitation section. We have followed the 63 patients identified in 2018, using the nursing documentation. 25 PUs healed during the follow up (40,3%).
Line 153 - how was randomization made?
Since most nurses worked in several units, each nurse was allocated to the unit type where the nurse worked more hours monthly. In the case of a tie, random allocation was used, implemented by the analytical software.
Table 4 - missing a headline with categories and themes emerged by the open question. Please specify how have you analyzed the content from this open question. Why ask this question if the content doesn't value the text?
We have analysed the content of this section by relating the issues raised by nurses to questions asked previously in NWI-R-PT. We have clarified it in the text and table.
Discussion - on your results, you report 38.4% of nurses with Education in Wound Care. Did you analyse if the PUs on the patients where those nurses worked had a better outcome? Did they have a better healing time?
We have analysed healing time per typology of unit.
For more details please see the revised version manuscript.
Round 2
Reviewer 1 Report
Dear Authors,
I have carefully read your revised research paper on the evaluation of the work environment in long-term care units and its association with wound care quality. Overall, your study provides valuable insights into the relationship between the work environment and patient outcomes in LTC units. However, I would like to offer some suggestions to further enhance the quality and impact of your research. Firstly, it would be beneficial to clearly articulate the research objective and hypothesis in the introduction section. This will provide a clear and focused direction for the study, enabling readers to better understand the purpose and significance of your research. Furthermore, I suggest presenting the key findings in a structured and organized manner. Consider grouping related findings together and using subheadings to improve readability. Additionally, compare and contrast your findings with previous literature to highlight the novelty and contribution of your study to the existing body of knowledge. In the discussion section, go beyond the presentation of results and delve into the practical implications of your findings for healthcare practice. Discuss how the identified associations between the work environment and wound care quality can inform interventions and improvements in LTC units. Providing specific recommendations for creating a more supportive work environment will add value to your research and offer actionable insights to stakeholders in the field. It is also important to address the limitations of your study. Discuss any potential sources of bias or confounding factors that may have influenced your results. Acknowledging these limitations demonstrates scientific rigor and helps readers interpret the findings appropriately. In the conclusion section, summarize the main findings concisely and reiterate the need for system-level changes in LTC units to foster supportive work environments. Emphasize the implications of your research for policy-makers, healthcare administrators, and nursing professionals, highlighting the urgency of addressing the current shortage of nurses and promoting positive work environments.
Ensure a smooth and logical flow throughout the paper. Proofread carefully for grammatical errors, typographical mistakes, and inconsistencies in writing style and formatting
Author Response
Dear Reviewer,
Thank you for your suggestions and the time taken to improve our article.
We have answered all your questions and comments.
Regards,
Kátia Furtado and co-authors.
-----------------------------------------------
Dear Authors,
I have carefully read your revised research paper on the evaluation of the work environment in long-term care units and its association with wound care quality. Overall, your study provides valuable insights into the relationship between the work environment and patient outcomes in LTC units. However, I would like to offer some suggestions to further enhance the quality and impact of your research.
Firstly, it would be beneficial to clearly articulate the research objective and hypothesis in the introduction section. This will provide a clear and focused direction for the study, enabling readers to better understand the purpose and significance of your research.
Response: We have added:
“It is part of a broad project to study barriers to the implementation of evidence-based practice related to wound care in LTC units, which is based on the premise that patient outcomes are influenced by how care services are organised and delivered. The organisation of care is a multidimensional concept influenced by knowledge, leadership, work environment and financial resources.”
Furthermore, I suggest presenting the key findings in a structured and organized manner. Consider grouping related findings together and using subheadings to improve readability. Additionally, compare your findings with previous literature to highlight the novelty and contribution of your study to the existing body of knowledge.
Response: We have added:
The NWI-R-PT questionnaire consists of three parts: Part A - Sociodemographic Characteristics; Part B - Characteristics of the nursing work environment most valued by Nurses.; Part C – Suggestions for improving practice. Results will be presented in this order”
We have added the following subheadings, as suggested.
3.1.1 Socio-demographic characteristics
3.1.2. Work Environment
3.1.3. Suggestions for improving practice
In the Discussion section, we compare our results with previous studies, but the evidence is poor.
In the discussion section, go beyond the presentation of results and delve into the practical implications of your findings for healthcare practice. Discuss how the identified associations between the work environment and wound care quality can inform interventions and improvements in LTC units. Providing specific recommendations for creating a more supportive work environment will add value to your research and offer actionable insights to stakeholders in the field.
Response: We have added in the final part of the discussion:
“Although there is no established way to organise care for pressure ulcer prevention and treatment, there are examples of provider-orientated interventions by introducing wound care champions figures, evidence-based care pathways and improvements in nursing documentation [78,79].“
It is also important to address the limitations of your study.
Discuss any potential sources of bias or confounding factors that may have influenced your results. Acknowledging these limitations demonstrates scientific rigor and helps readers interpret the findings appropriately.
Response: It is written in the limitations section:
The number of pressure ulcers followed was low, and the records were inaccurate, resulting in limited statistical power, which could be a potential reason for the weak correlation between the nursing work environment and PUs' quality of care. Besides this, the information on the nurses' working environment and wound healing came from two separate studies, and most of the nurses who answered the NWI-R-PT were there for no more than five years.
Response: We have added:
The available confounder candidates were a limited set of risk factors for PUs, and we could not take other possible confounders into account, such as incontinence, lack of sensory perception, obesity, poor nutrition and hydration, and heart failure, renal disease and mental illnesses such as Alzheimer's disease.
In the conclusion section, summarize the main findings concisely and reiterate the need for system-level changes in LTC units to foster supportive work environments. Emphasize the implications of your research for policy-makers, healthcare administrators, and nursing professionals, highlighting the urgency of addressing the current shortage of nurses and promoting positive work environments.
Response: We have added the following:
“In general, nurses are satisfied with their work environment in long-term care units in Alentejo. We found low-quality and inaccurate nursing documentation related to wound care and observed low healing rates. Age, immobility, and the presence of more than two risk factors impaired healing. Good distribution of nursing staff over the units might improve the quality of wound care, but the other studied work environment factors did not impact the quality of care.”
Comments on the Quality of English Language
Ensure a smooth and logical flow throughout the paper. Proofread carefully for grammatical errors, typographical mistakes, and inconsistencies in writing style and formatting.
Response: We have revised the text where needed.